# Effects of Action Observation Plus Motor Imagery Administered by Immersive Virtual Reality on Hand Dexterity in Healthy Subjects

**DOI:** 10.3390/bioengineering11040398

**Published:** 2024-04-19

**Authors:** Paola Adamo, Gianluca Longhi, Federico Temporiti, Giorgia Marino, Emilia Scalona, Maddalena Fabbri-Destro, Pietro Avanzini, Roberto Gatti

**Affiliations:** 1Physiotherapy Unit, IRCCS Humanitas Research Hospital, Via Manzoni 56, 20089 Rozzano, Milan, Italy; 2Department of Biomedical Sciences, Humanitas University, Via Rita Levi Montalcini 4, 20072 Pieve Emanuele, Milan, Italy; 3Dipartimento di Scienze Medico Chirurgiche, Scienze Radiologiche e Sanità Pubblica (DSMC), Università Degli Studi di Brescia, Viale Europa 11, 25123 Brescia, Brescia, Italy; 4Consiglio Nazionale Delle Ricerche, Istituto di Neuroscienze, Via Volturno, 39-E, 43125 Parma, Parma, Italy

**Keywords:** action observation, motor imagery, virtual reality, manual dexterity

## Abstract

Action observation and motor imagery (AOMI) are commonly delivered through a laptop screen. Immersive virtual reality (VR) may enhance the observer’s embodiment, a factor that may boost AOMI effects. The study aimed to investigate the effects on manual dexterity of AOMI delivered through immersive VR compared to AOMI administered through a laptop. To evaluate whether VR can enhance the effects of AOMI, forty-five young volunteers were enrolled and randomly assigned to the VR-AOMI group, who underwent AOMI through immersive VR, the AOMI group, who underwent AOMI through a laptop screen, or the control group, who observed landscape video clips. All participants underwent a 5-day treatment, consisting of 12 min per day. We investigated between and within-group differences after treatments relative to functional manual dexterity tasks using the Purdue Pegboard Test (PPT). This test included right hand (R), left hand (L), both hands (B), R + L + B, and assembly tasks. Additionally, we analyzed kinematics parameters including total and sub-phase duration, peak and mean velocity, and normalized jerk, during the Nine-Hole Peg Test to examine whether changes in functional scores may also occur through specific kinematic patterns. Participants were assessed at baseline (T0), after the first training session (T1), and at the end of training (T2). A significant time by group interaction and time effects were found for PPT, where both VR-AOMI and AOMI groups improved at the end of training. Larger PPT-L task improvements were found in the VR-AOMI group (*d:* 0.84, CI_95_: 0.09–1.58) compared to the AOMI group from T0 to T1. Immersive VR used for the delivery of AOMI speeded up hand dexterity improvements.

## 1. Introduction

Cognitive facilitations including action observation (AO) and motor imagery (MI) have been recently adopted to promote motor learning in athletes or in subjects with motor impairments [1,2].

AO entails the observation of motor tasks through video clips, which are delivered using a laptop screen showing motor contents representing motor acts executed by age and gender-matched subjects [1,3,4]. This stimulation recruits the mirror neuron system (MNS), a frontoparietal network active both during the execution and observation of motor acts. The MNS plays a key role in understanding actions performed by others [5] and is involved in the building of motor memories related to the observed tasks [6]. The neural solicitation driven by AO is dependent on the features of the observed stimuli, such as the observer perspective or the transitive or intransitive nature of the action. For example, a first-person perspective induces higher MNS activation during the observation of upper limb transitive actions, compared to a third-person perspective [7,8]. In addition, the impact of AO on motor learning is larger when the observed actions are accompanied by action-related sounds [9].

MI consists of a cognitive process in which subjects imagine themselves executing motor tasks without performing them [10,11]. MI performed immediately after AO has been reported to further enhance motor learning when compared to the execution of AO alone [11]. The rationale behind this finding lies in the overlap between the neural substrates of AO and MI, both of which share large territories over frontoparietal circuits [10]. In fact, the association of AO and MI (AOMI) has been adopted to enhance the performance of sport-related gestures [12]. Moreover, AOMI-related benefits have been demonstrated in terms of motor re-learning in patients with motor impairments [13] when compared to the administration of AO or MI alone.

When considering treatments integrating action observation (e.g., AOMI), an additional resource is represented by virtual reality (VR), which is a computer-generated simulation of a three-dimensional image or environment that users can interact with in a seemingly real or physical manner. This interaction is facilitated through specialized devices such as helmets with a screen inside or controllers fitted with sensors [14,15]. VR technologies may be “immersive” when the subject is completely immersed in the virtual environment and visual information change according to the movement of the user’s head [16]. Immersive VR offers multiple advantages. On one hand, it leads to the illusion of being in a virtual place and the first-person perspective generates the illusion that the virtual body is the observer’s own [17]. The observed upper limbs are coherent with the real limbs of the subject, enhancing the embodiment of the observer [18], the sense of agency, the self-attribution of the virtual body [19,20], and the overall engagement during the treatment [21]. On the other hand, neuroimaging studies have proven that action observation presented in 3D evokes a greater sensorimotor response than 2D stimuli [22]. All these advantages induced clinical researchers to adopt immersive VR in the field of motor rehabilitation, demonstrating its efficacy in favoring the upper limb recovery in patients with neurological disorders [23,24]. In this scenario, previous studies have suggested that implicit motor learning, defined as the acquisition of new motor skills without the awareness of the learning process, may be enhanced when subjects are embodied in immersive virtual environments and when they have a first-person perspective [25]. The embodiment with the virtual body might increase AO effects, since the mirror mechanism has been reported as modulated not only by the observed movement, but also by action meaning and environmental, contextual, and emotional factors [26,27,28].

Few studies have described the link among AO, VR, and MI in terms of motor learning. Choi [29] described higher motor imagery abilities following hand movements observed in immersive VR. Moreover, the administration of AO through immersive VR to patients in a subacute phase after a stroke has shown improvements in upper limb motor function when compared to upper limb motor rehabilitation performed without VR [30]. It is worth noting that the aforementioned cognitive approaches have been frequently associated with exercises aimed at improving hand dexterity [31,32]. Manual dexterity consists of the ability to execute coordinated hand and fingers movements and derives from the integration of hand biomechanics with sensorimotor and cognitive processes [32]. When considering hand function, improvements in manual dexterity after training represent an index of enhanced motor control during hand and fingers movements [33]. However, although literature data suggest the opportunity to enhance the embodiment through the use of immersive VR systems, no studies have investigated the effect of an AOMI training administered through immersive VR on manual dexterity when compared to a conventional AOMI training.

Therefore, the aim of this study was to investigate the effects on manual dexterity of AOMI delivered through immersive VR compared to conventional AOMI delivered via a laptop screen in healthy subjects. The study hypothesis was that immersive VR might boost the effects of AOMI on motor learning in healthy subjects.

This modality of administration may increase motor resonance, since AOMI is modulated by contextual and environmental factors associated with observed movement [26,28]. The novelty of the study lies in leveraging the embodiment induced by action observation and motor imagery delivered using immersive VR, alongside the first-person perspective, rather than via a laptop screen to induce motor learning in terms of manual dexterity changes.

To measure the extent and changes in manual dexterity, we identified specific functional tests that were administered both before and after a week of daily training. However, motor learning may manifest as an improvement in a functional test, as well as a higher velocity at which this improvement occurs [34]. For this reason, the functional battery was administered also after the first training session. Additionally, we investigated the movement kinematics during manual dexterity tasks to examine whether changes in functional scores may also occur through specific kinematic patterns.

We provide an outline of the manuscript to guide readers through its content. In the Background section, we highlight the rationale behind why VR may boost AOMI effects on motor learning. Subsequently, in the Methods section, we describe the study design, the sample included in the study, the different characteristics of treatment administered, and a detailed description of the outcome measures chosen to assess manual dexterity. The Results section presents the effects of different treatments, whose explanations, interpretations, and implications are addressed in the Discussion and Conclusions.

## 2. Materials and Methods

### 2.1. Participants

The sample size was estimated based on previous reports about the Purdue Pegboard Test ([35], see below). It was estimated that, considering an alpha error of 5%, a minimum of 15 participants would be required in each group to provide 80% power to detect a Cohen’s *d*  = 1.0 (large effect size) between VR-AOMI and AOMI groups at T2 [36].

Forty-five healthy subjects (15 females, 30 males; mean age 23.4 ± 2.68 years) were enrolled, and their characteristics are reported in Table 1. Inclusion criteria were: (1) age between 20 and 35 years, (2) right-handedness according to the Edinburgh Handedness Inventory [37]. Conversely, the presence of upper limb sensorimotor disorders or recent traumatic injuries, the usual performance of motor activities or sports involving remarkable manual skills (e.g., playing instruments, juggling), a history of epilepsy seizures, and visual impairments that cannot be corrected with lenses were used as exclusion criteria. All participants signed a written informed consent and the study protocol was approved by the Ethical Committee of the Humanitas Clinical and Research Center (approval number: VR-AOT-GR-2019).

### 2.2. Intervention

Action stimulus depicted a humanoid avatar playing a pianola with the left hand from a first-person perspective (Figure 1). The action performed by the humanoid avatar replicated the kinematics recorded from a healthy subject [38] by a motion capture system (Awinda, XSens, Enschede, The Netherlands) incorporating industrial gloves to track hand and finger movements (Manus Prime II Xsens, Enschede, The Netherlands). Given the immersive nature of the stimuli, VR-AOMI participants wearing the headset could explore the surrounding space by moving head and gaze accordingly.

Both VR-AOMI and AOMI participants were asked to carefully observe the action for 3 min, with an audio trace coherent with piano keys pressure. After the observation, they had to imagine themselves performing the action (MI) for one minute, avoiding any active movement. Observation and imagination were repeated three times (12 min in total) for five consecutive days, maintaining constant the daytime of task administration. Finally, participants of the CTRL group observed landscapes (free of any biological motor contents) in immersive VR for the same amount of time as the AOMI groups.

### 2.3. Study Design, Randomization, and Enrollment

The study has a three-armed, single-blind, randomized, controlled design. Participants were recruited by an independent researcher not involved in the subsequent stages of the study and they were assigned to one of the three experimental groups according to a random computer-generated list. The VR-AOMI group (15 subjects) underwent AOMI through immersive presentation of the action stimuli, the AOMI group (15 subjects) observed the same stimuli via a laptop screen, while the CTRL group (15 subjects) observed landscape videos via the VR visor (Oculus II).

### 2.4. Functional and Kinematic Assessment

The kinesthetic and visual imagery questionnaire (KVIQ) was administered to participants at baseline to assess their motor imagery abilities [39].

The main outcome of the study pertained to the participants’ hand dexterity. For this reason, all subjects underwent a functional and kinematic assessment at baseline (T0), after the first training session (T1—day 1), and at the end of training (T2—day 5). The study timeline is shown in Figure 2.

The assessment procedures were conducted by a researcher unaware of group allocation at the Motion Analysis Lab of the Humanitas Clinical Institute, Milan, Italy.

The functional assessment consisted of the Purdue Pegboard Test (PPT), including four subtests: PPT-R (right hand), PPT-L (left hand), PPT-B (both hands simultaneously), and an assembly task. Participants had 30 s for each peg-insertion task and 60 s for the assembly task [35].

The kinematic assessment included the Nine-hole peg test (NHPT) [40] and the finger tapping test (FTT) [41]. During the NHPT, participants were seated on a height-adjustable chair with the pegboard positioned on a table in front of them. They were asked to grasp the pegs from a container one by one, place them into a nine-hole board, remove the pegs from the board, and replace them in the container as quickly as possible [40]. Kinematic data were recorded using an optoelectronic system (SMART-DX, BTS, Milan, Italy) equipped with eight infrared cameras and 16 reflective markers, of which three were on the table to define the global reference system and 13 were on anatomical landmarks. The system calibration involved a 10 s static test with four additional markers on the NHPT board.

Markers trajectories were filtered using a fourth-order low-pass Butterworth filter (cut-off 4 Hz). Subsequently, we extracted the total and single-phase times (peg grasp, peg transfer, peg in hole, hand return), normalized jerk, mean and peak velocity during peg transfer and hand return phases [40]. Subjects performed two trials for both the right and the left sides, and the shortest one in terms of total time of execution was considered for the analysis.

Kinematic assessment also included the finger tapping test (FTT), performed with participants seated on a chair, forearms resting on a desk with elbows at 90 degrees flexion. They performed a tapping sequence (thumb, index, middle, ring, and little finger) at maximum speed for 15 s without visual feedback. Fingers were marked distally with smaller reflective markers (6 mm diameter). Errors were recorded for incorrect sequence movements. The total number of errors and the total number of movements considering all the fingers were analyzed [41].

### 2.5. Statistical Analysis

After verifying the normality assumption through the Shapiro–Wilk test, a parametric pipeline was adopted. Univariate ANOVA and chi-square test were used to investigate between-group differences in terms of demographic characteristics and KVIQ at baseline. A 3 × 3 mixed ANOVA with time as within-subjects and group as between-subjects factors was used to evaluate differences in terms of manual dexterity assessed by functional test and kinematic analysis between groups over time. Post-hoc analyses were Bonferroni-corrected to reduce the false positive ratio. In addition, changes from baseline to post-treatment (deltas) were calculated and the effect size (Cohen’s *d*), with its 95% confidence interval (CI_95_), was also computed between deltas in the three groups and between each time point in the same group and interpreted as small (0.2 ˂ *d* ˂ 0.5), medium (0.5 ˂ *d* ˂ 0.8), large (0.8 ˂ *d* ˂ 1.3), and very large (*d* > 1.3). Analyses were conducted using SPSS Statistics 29.0 for iOS and the statistical level of significance was set to α = 0.05.

## 3. Results

None of the participants withdrew from the study and no between-group differences were found in terms of baseline characteristics including age, weight, height, gender, and KVIQ score, as shown in Table 1.

### 3.1. The Effects of AOMI Applied through VR on Manual Dexterity Assessed with the Purdue Pegboard Test

A significant time by group interaction and time effect were found for PPT-R, PPT-L, PPT-R + L + B, and PPT-Assembly tasks, while a time effect was found for PPT-Both.

The graphs in Figure 3 show the PPT scores obtained across the three groups, while numeric values for within- and between-group comparisons of deltas are reported in Table 2 and Table 3, respectively.

Table 2 reports within-group differences expressed as Cohen’s d and CI_95_. Both VR-AOMI and AOMI groups improved from T0 to T1 and further experienced enhanced manual dexterity from T0 to T2 during PPT-R, PPT-L, PPT-Both, PPT-R + L + B, and PPT-Assembly tasks. These results suggest a major effect driven by action observation regardless of its visual display features. The CTRL group improved from T0 to T2 only for PPT-Both, PPT-R + L + B, and PPT-Assembly tasks.

Table 3 reports between-group differences expressed as Cohen’s *d* and CI_95._ Differences between VR-AOMI and AOMI groups in terms of deltas were exclusively found for PPT-L at T1 in favor of VR-AOMI (*p* = 0.029, *d:* 0.84, *CI_95_*: 0.09–1.58), indicating that the use of virtual reality speeded up the motor learning process, reaching a significant increase in dexterity already after the first training session (Figure 4). In addition, between-group differences in terms of deltas (T1–T0 and T2–T0) were detected in favor of both AOMI groups compared to the CTRL group in all subtasks of PPT (Table 3).

### 3.2. The Effects of AOMI Applied through VR on Kinematic Assessment during the Nine-Hole Peg Test

Results for the NHPT with the left hand revealed a time effect for removing time (*p* = 0.003), peak return velocity (*p* = 0.009), transfer time (*p* = 0.040), return time (*p* = 0.029), and transfer velocity (*p* = 0.042) (Appendix A).

Within-group post-hoc analyses revealed a decrease in removing time in the AOMI group at T2 compared to T0 (*MD:* 0.93, *p* = 0.041, *CI_95_:* 0.03, 1.83), while the VR-AOMI group revealed a decrease in transfer time at T2 when compared to T0 (*MD:* 0.38, *p* = 0.05, *CI_95_:* 0.76). The CTRL group revealed a decrease in return time at T2 compared to T1 (*MD:* 0.31, *p* = 0.013, *CI_95_:* 0.05, 0.57). Both the AOMI (*MD:* −0.11, *p* = 0.04, *CI_95_:* −0.19, 0.03) and control groups (*MD:* −0.09, *p* = 0.012, *CI_95_*: −0.18, −0.02) achieved higher peak velocity during the return phase at T2 compared to T1.

The NHPT performed with the right hand revealed a time effect for removing time (*p* = 0.003) and a time by group interaction for total test (*p* = 0.025) and peg-in-hole times (*p* = 0.043). Between-group post-hoc analyses revealed a lower peg-in-hole time in the CTRL group compared to the AOMI group at T2 (*MD:* 0.76, *p* = 0.036, *CI_95_:* 0.04, 1.48). Finally, both VR-AOMI (*MD:* 0.47, *p* = 0.027, *CI_95_:* 0.06, 0.89) and CTRL (*MD:* 0.46, *p* = 0.030, *CI_95_:* 0.05, 0.88) groups showed a decrease in terms of removing time from T0 to T2 (Appendix A).

### 3.3. The Effects of AOMI Applied through VR on Kinematic Assessment during the Finger Tapping Test

A time effect was found in total errors in the FTT performed with the left hand (*p* = 0.002). Post-hoc analyses revealed improvement in the AOMI group (*p* = 0.027) at T2 compared to T0. A time effect was found also in total fingers movement (*p* < 0.001). Post-hoc analyses revealed an improvement only in the AOMI group (*p* = 0.003) from T1 to T0, while all the groups showed an improvement (*p* < 0.001) at T2 compared to T0. Only the VR-AOMI group improved (*p* = 0.025) from T1 to T2. No improvements were found during the FTT with the right hand.

## 4. Discussion

This was the first study aimed at investigating whether immersive VR boosts the effects of AOMI. The main study finding was that AOMI delivered via immersive VR speeded up improvements in manual dexterity in healthy subjects. In fact, although AOMI groups showed similar manual dexterity abilities at the end of training, the VR-AOMI group achieved significantly greater dexterity changes than the AOMI group after the first training session. On the other hand, the AOMI group gradually improved manual dexterity over the five training sessions. Finding the most effective strategies to fasten acquisition of motor skills is fundamental for motor learning relative to both athletes and patients undergoing motor rehabilitation, where one goal is reducing the recovery time of subjects undergoing rehabilitation plannings.

Our study applied this approach to hand dexterity, since the effects of AOMI on the upper limbs are the most investigated in the literature and are described as the most promising in motor rehabilitation [13]. The choice to investigate healthy subjects derived from the desire to avoid any discomfort in patients assigned to the VR-AOMI group, since previous studies have described cybersickness as a potential side effect during the use of immersive virtual reality systems [42]. However, few studies have reported minimal cybersickness using a head-mounted display in healthy subjects and stroke patients during upper limb exercises through fully immersive VR systems [43]. Finally, the choice to associate AO and MI derives from literature data supporting the efficacy of AOMI in promoting the learning of complex motor tasks when compared to the use of AO or MI alone in healthy subjects [44].

The current study demonstrates that virtual reality speeded up motor learning induced by AOMI, as reported by PPT—Left hand and assembly tasks results. Based on these findings, immersive virtual reality may have enhanced embodiment and amplified 3D kinematics details of the motor task, boosting the acquisition of complex motor skills in young healthy subjects [1]. Interestingly, the largest effect in both AOMI groups was found for left ‘trained’ hand, consistently with previous studies which have demonstrated that motor learning is related to the characteristics of the observed movement [45]. Improvements at the level of the right untrained hand may derive from an interlimb transfer effect from the non-dominant to the dominant hand after a unilateral dexterity training [46].

Nevertheless, a certain heterogeneity in terms of the PPT-Assembly task was found at baseline, where the control group revealed higher mean score than the AOMI groups. However, this heterogeneity may be attributed to higher variability in terms of assembly task performance compared to unimanual tasks during PPT execution in healthy subjects [35]. The opportunity to speed up the effects of AOMI through its association with VR may depend on a greater sense of embodiment, since the motor learning process has been shown to increase when embodiment with the observed movements is higher [15]. In fact, resonance of MNS does not depend only on observed actions, since previous studies have suggested that environment and context factors also modulate corticospinal excitability during AO [26,28]. Particularly, the inferior frontal gyrus and the ventral premotor cortex revealed a significant signal increase when the context suggested the intention associated with hands actions [26]. Familiarity with and expertise of the observed action has also been shown to increase the MNS resonance [47], and the embodiment induced by immersive VR may increase the perception of familiarity with the observed action [48].

Moreover, stereoscopic 3D stimuli may provide a greater number of kinematic details than bidimensional visual stimuli, allowing for a precise assessment and understanding of the observed actions [22,49]. Finally, it is worth noting that the sense of embodiment has been also described as influenced by the person-related perspective, since an egocentric perspective has been demonstrated to trigger the illusion of body ownership and self-attribution of the virtual body [50]. Overall, VR-AO delivered in the first-person-perspective may also be considered as a multisensory stimulation which exploits the simultaneous use of different sensory inputs to enhance motor learning processes [51,52]. In particular, immersive VR systems allow for the integration of realistic scenarios engaging the patient’s sensorimotor system and promoting the feeling of being into the virtual environment [53]. In this scenario, the current study findings induce us to consider AO, MI, and VR as mutually beneficial. In fact, immersive VR might enhance motor imagery performance, providing a rich immersive and illusive experience especially in first-person perspective virtual scenarios [29,54].

The current study found improvements after AO treatments in PPT, while differences in the 9HPT and finger tapping test were not detected with respect to AO or CTRL groups. PPT has been previously described as more sensitive than the nine-hole peg test in detecting changes in manual dexterity, especially in healthy subjects [55]. Little improvements in the finger tapping test and bimanual tasks were found in the control group, probably as a result of a learning effect caused by the repetition of the test three times in one week [56].

Although VR seems to speed up improvements in manual dexterity specifically in the left observed hand, changes at the end of the treatment period were the same for immersive and non-immersive AOMI. However, confirmation of this result in clinical practice would have an interesting relevance, mostly in the rehabilitation field, where faster functional recovery is a goal of rehabilitation [57].

## 5. Conclusions

In conclusion, the results of the study suggest that AOMI delivered through immersive VR speeded up hand dexterity improvements. The study has some limitations. First, a single exercise was delivered during the treatment period and improvements may have been greater with a progression of exercise difficulty within the treatment period. This limitation derived from the fact that exercises recorded in virtual reality were specifically designed for the rehabilitation of post-stroke patients in a subacute phase. Thus, the authors chose the most challenging task for a healthy young subject. Furthermore, visually induced motion sickness was not explored, which has been described as a possible side effect of VR immersion, with symptoms including nausea, disorientation, and oculomotor discomfort [38]. However, no dropouts were reported either in the landscape or the VR-AOMI groups. Finally, the study was unbalanced in terms of gender and age according to the inclusion criteria, as the number of males was twice that of females, and the mean age was lower with respect to the mean age computed from the specified inclusion criteria. However, the proportion of males and females in the VR-AOMI, AOMI, and CTRL groups, as well as the mean age in all three groups, did not show statistically significant differences.

Further studies are needed to confirm the current findings in clinical practice and explore the opportunity to reduce the recovery time in subjects undergoing rehabilitation programs.

## Figures and Tables

**Figure 1 bioengineering-11-00398-f001:**
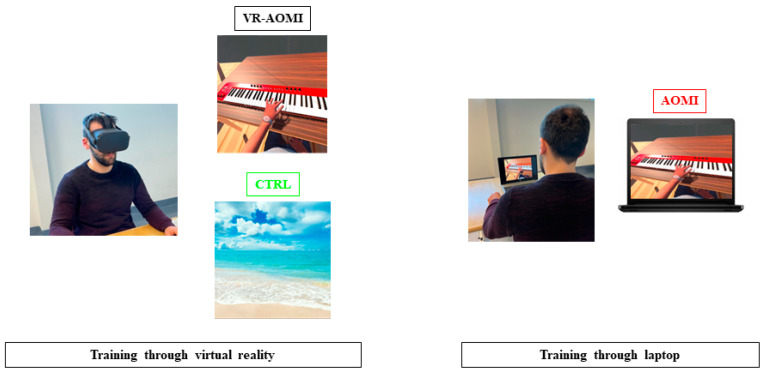
Study participant undergoing the VR-AOMI through the Oculus headset (VR-AOMI), AOMI through a laptop screen (AOMI), and landscape observation through VR (CTRL).

**Figure 2 bioengineering-11-00398-f002:**
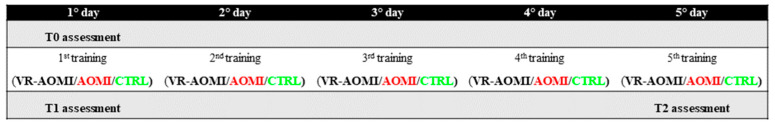
Study timeline.

**Figure 3 bioengineering-11-00398-f003:**
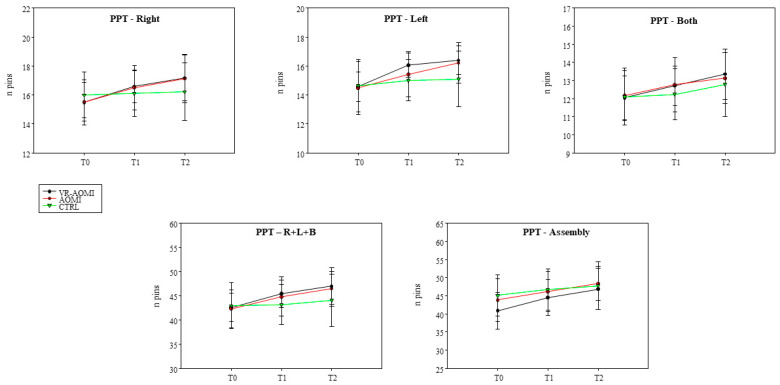
The scores obtained in Purdue Pegboard Test across the three groups are shown. Data are presented as means (dots and triangles) and standard deviation (vertical bars).

**Figure 4 bioengineering-11-00398-f004:**
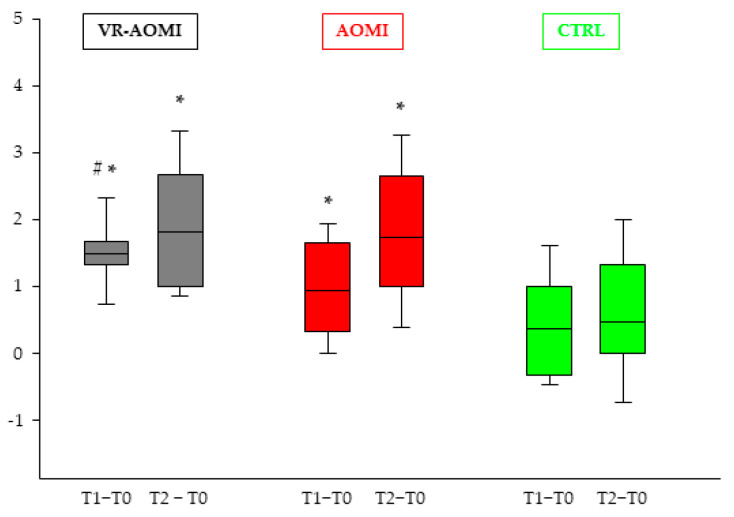
Between-group differences for deltas from T0 to T1 and from T2 to T0 for the PPT-L task. Boxes represent the range between the first and the third quartile, the middle horizontal line is the mean value, and the ends of the vertical line, from top to bottom, are the maximum and minimum values, respectively. Symbols show differences in terms of delta between the VR-AOMI and AOMI groups from T0 to T1 (#) and between the VR-AOMI and AOMI groups with the CTRL group from T0 to T1 and from T0 to T2 (*).

**Table 1 bioengineering-11-00398-t001:** Characteristics of study participants. Data are shown as mean ± standard deviation.

	VR-AOMI Group	AOMI Group	CTRL Group	*p*-Value
	(n = 15)	(n = 15)	(n = 15)	
**Age (years)**	24.06 ± 3.1	23.53 ± 3.24	22.53 ± 1.06	0.32
**Weight (kg)**	67.43 ± 10.82	66.06 ± 12.52	69.53 ± 8.71	0.67
**Height (cm)**	176.53 ± 6.76	172.6 ± 11.09	172.73 ± 6.96	0.39
**Gender**	11M/4F (27%F)	9M/6F (40%F)	10M/5F (33%F)	0.74
**KVIQ**	36.6 ± 7.22	40.87 ± 6.42	38.27 ± 8.13	0.35

**VR-AOMI**: action observation and motor imagery through immersive VR; **AOMI**: action observation and motor imagery through laptop; **CTRL**: control; **KVIQ**: kinesthetic and visual imagery questionnaire; **M:** male; **F**: female.

**Table 2 bioengineering-11-00398-t002:** PPT scores in the VR-AOMI (**A**), AOMI (**B**), and CTRL (**C**) groups at T0, T1, and T2 are expressed as mean ± standard deviation. Within-group comparisons between T1/T0 and T2/T0 are expressed as Cohen’s *d* with 95% confidence interval (CI_95_).

(**A**)					
**VR-AOMI**	**T0**	**T1**	**T2**	***d* (CI_95_)** **T1–T0**	***d* (CI_95_)** **T2–T0**
**R task**	15.49 ± 1.57	16.60 ± 1.14	17.17 ± 1.57	1.03 (0.39, 1.65)	1.51 (0.75, 2.52)
**L task**	14.58 ± 1.03	16.06 ± 0.85	16.40 ± 0.99	2.37 (1.53, 3.36)	1.84 (0.99, 2.67)
**B Task**	12.04 ± 1.22	12.71 ± 1.09	13.35 ± 1.38	0.75 (0.16, 1.31)	1.79 (0.95,2.61)
**R + L + B task**	42.62 ± 2.94	45.42 ± 2.82	46.98 ± 3.80	1.75 (0.92, 2.56)	2.07 (1.15, 2.97)
**Assembly task**	40.86 ± 5.02	44.55 ± 4.94	46.86 ± 5.70	1.86 (0.99, 2.69)	2.73 (1.60, 3.85)
(**B**)					
**AOMI**	**T0**	**T1**	**T2**	***d* (CI_95_)** **T1–T0**	***d* (CI_95_)** **T2–T0**
**R task**	15.53 ± 1.33	16.51 ± 1.53	17.13 ± 1.67	0.98 (0.34,1.58)	1.51 (0.74, 2.42)
**L task**	14.49 ± 1.83	15.42 ± 1.55	16.22 ± 1.38	1.35 (0.63, 2.05)	1.70 (0.89, 2.50)
**B Task**	12.18 ± 1.37	12.77 ± 1.50	13.13 ± 1.41	0.84 (0.24, 1.42)	1.09 (0.43, 1.72)
**R + L + B task**	42.33 ± 3.94	44.29 ± 3.90	46.46 ± 3.56	1.38 (0.65, 2.08)	1.91 (1.03, 2.76)
**Assembly task**	43.86 ± 5.91	46.15 ± 5.58	48.40 ± 4.68	0.62 (0.06, 1.17)	1.11 (0.45, 1.75)
(**C**)					
**CTRL**	**T0**	**T1**	**T2**	***d* (CI_95_)** **T1–T0**	***d* (CI_95_)** **T2–T0**
**R task**	16.00 ± 1.58	16.10 ± 1.57	16.22 ± 2.00	0.1 (−0.41, 0.60)	0.22 (−0.30, 0.73)
**L task**	14.64 ± 1.81	15.02 ± 1.43	15.11 ± 1.92	0.52 (−0.3, 1.05)	0.50 (−0.04, 1.04)
**B Task**	12.11 ± 1.58	12.24 ± 1.40	12.80 ± 1.77	0.16 (−0.35, 0.67)	1.18 (0.54, 1.84)
**R + L + B task**	42.95 ± 4.73	43.15 ± 4.18	44.11 ± 5.42	0.09 (−0.42, 0.60)	0.72 (0.14, 1.28)
**Assembly task**	45.09 ± 5.77	46.78 ± 5.69	47.78 ± 6.53	0.90 (0.28, 1.49)	1.53 (0.76, 2.28)

**Abbreviations. VR-AOMI**: action observation through immersive virtual reality group; **AOMI**: action observation group; **CTRL**: control group; ***d***: Cohen’s *d*; **CI_95_**: confidence interval; **R**: right; **L**: left; **B**: both.

**Table 3 bioengineering-11-00398-t003:** Comparisons of deltas among the VR-AOMI, AOMI, and CTRL groups for the Purdue Pegboard Test. Data were expressed as Cohen’s *d* with 95% confidence interval (CI_95_). Significant results are shown in bold text.

	ΔT1–T0 Cohen’s *d* (CI_95_)			ΔT2–T0 Cohen’s *d* (CI_95_)		
	VR-AOMI/AOMI	VR-AOMI/CTRL	AOMI/CTRL	VR-AOMI/AOMI	VR-AOMI/CTRL	AOMI/CTRL
**R task**	0.13 (−0.59–0.84)	**0.90** **(0.14–1.65)**	**0.81** **(0.06–1.55)**	0.08 (−0.63–0.80)	**1.39** **(0.58–2.19)**	**1.35** **(0.54–2.13)**
**L task**	**0.84** **(0.09–1.58)**	**1.64** **(0.79–2.46)**	**0.79** **(0.04–1.52)**	0.09 (−0.63–0.80)	**1.42** **(0.60–2.21)**	**1.31** **(0.50–2.09)**
**B Task**	0.08 (−0.63–0.80)	0.62 (−0.12–1.35)	0.60 (−0.14–1.33)	0.44 (−0.29–1.16)	**0.94** **(0.18–1.69)**	0.36 (−0.37–1.08)
**R + L + B task**	0.18 (−0.54–0.90)	**1.38** **(0.57–2.17)**	**1.16** **(0.38–1.93)**	0.09 (−0.62–0.81)	**1.71** **(0.85–2.54)**	**1.57** **(0.73–2.38)**
**Assembly task**	0.48 (−0.26–1.20)	**1.03** **(0.26–1.79)**	0.21 (−0.51–0.92)	0.45 (−0.28–1.17)	**1.67** **(0.82–2.49)**	0.59 (−0.15–1.32)

**Abbreviations. VR-AOMI**: action observation performed through immersive virtual reality group; **AOMI**: action observation group; **CTRL**: control group; **CI_95_**: confidence interval; **R**: right; **L**: left; **B**: both.

## Data Availability

The dataset used and analyzed during the current study is available from the corresponding author upon reasonable request.

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
