# Peer review of "Effects of Action Observation Plus Motor Imagery Administered by Immersive Virtual Reality on Hand Dexterity in Healthy Subjects"

_bioengineering, 2024, doi:10.3390/bioengineering11040398_

Round 1
Reviewer 1 Report
Comments and Suggestions for Authors
I examined your paper titled "Effects of action observation plus motor imagery administered by immersive virtual reality on hand dexterity in healthy subjects" in detail. I have listed the points I found missing in the paper. Many expressions are written in parentheses in the Abstract section. These tire the reader. In the abstract section, it should be highlighted why the article was written, which methods were used, and what the results were. A paragraph regarding the organization of the article should be added at the end of the Introduction section. Additionally, a paragraph about the innovations in the article and its contributions to the literature should be added at the end of the Introduction section. Similar studies on the subject should be examined. It is very difficult to understand the method used in the paper. The method used in the study should be expressed in a figure and detailed. In the Result section, there is a lot of numerical data written in parentheses. It is very difficult to read and analyze these results. It is possible to present the results obtained as a result of examining the effects of action observation plus motor images applied by immersive virtual reality on dexterity in healthy subjects. The number of Figures and Tables in the study is almost non-existent. This is very important for the article to attract the reader's attention. Similar studies conducted in the Discussion section can be presented in a table and discussed. I would also like to point out that the similarity rate in the article is very high.
Comments on the Quality of English LanguageSpelling and grammatical errors should be reviewed.
Author Response
The authors express their gratitude to the Editor and the Reviewers for their valuable comments aimed at enhancing the quality of the study and for providing the opportunity to revise the manuscript.
The revised text is titled “Manuscript with track changes” and the sequence of revisions follows the sequence of requested changes.
# Reviewer 1
I examined your paper titled "Effects of action observation plus motor imagery administered by immersive virtual reality on hand dexterity in healthy subjects" in detail. I have listed the points I found missing in the paper.
Question: Many expressions are written in parentheses in the Abstract section. These tire the reader.
Answer: We thank the Reviewer for the comment. We removed most of the parentheses throughout the abstract to enhance the text's fluidity and clarity, retaining only a few for clarity, particularly concerning the acronyms of the three groups and assessment timepoints.
Question: In the abstract section, it should be highlighted why the article was written, which methods were used, and what the results were.
Answer: We thank the Reviewer for the comment. We revised the abstract to highlight the rationale (lines 17-20), the methodology employed for the study (lines 22-30) and the results, particularly focusing on differences in manual dexterity between groups differences before and after the VR-AOMI, AOMI and CTRL treatments.
Question: A paragraph regarding the organization of the article should be added at the end of the Introduction section.
Answer: We thank the Reviewer for the comment. We added a paragraph with an explanation of the organization of the manuscript at the end of Introduction section. We provided a brief explanation of each section of the paper to facilitate readers in navigating the content and highlight the relevance of the article (lines 132-139).
Question: Additionally, a paragraph about the innovations in the article and its contributions to the literature should be added at the end of the Introduction section. Similar studies on the subject should be examined.
Answer: We thank the Reviewer for the comment. We added a paragraph at the end of the Introduction section (lines 115-121) about the novelty of the study and its contribution to the literature. Previous paragraphs in this section referred to previous studies that have examined factors influencing motor learning, specifically those related to embodiment aimed at facilitating motor learning. Moreover, we specified the relevance and the rationale behind the choice of functional and instrumental manual dexterity assessments (lines 125-131).
Question: It is very difficult to understand the method used in the paper. The method used in the study should be expressed in a figure and detailed.
Answer: We thank the Reviewer for the constructive comment. Firstly, we reorganized the paragraph structure and revised the main text to enhance the clarity of the Methods section. Moreover, we added Figure 2 to explain the methodology used in the paper, outlining the timeline of assessments in relation to the five-day treatment period. Additionally, we modified Figure 1 to show the treatment description across the three groups.
Question: In the Result section, there is a lot of numerical data written in parentheses. It is very difficult to read and analyze these results. It is possible to present the results obtained as a result of examining the effects of action observation plus motor images applied by immersive virtual reality on dexterity in healthy subjects.
Answer: We thank the Reviewer for the comment. We decided to remove in the Results section a part of the text containing extensive numerical data within parentheses. Instead, we included these data in an additional table in the Results section (new Table 2) to make the reading and understanding of the main results more effortless, in accordance with the suggestion provided in the subsequent comment to incorporate more tables.
Additionally, we have revised the subtitles of the Results section to express the study findings in terms of the effects of AOMI delivered via VR on manual dexterity changes.
Question: The number of Figures and Tables in the study is almost non-existent. This is very important for the article to attract the reader's attention.
Answer: We thank the Reviewer for the comment. We added Figure 1, showing the treatment performed in VR-AOMI, AOMI and CTRL groups in the Method section, and Figure 2, describing the timeline of the study. Additionally, we have modified the Results section also adding Figure 4, showing the mail finding of the study (box plot with between-group comparison for deltas T1-T0 and T2-T0 for PPT-L task).
Moreover, we added Table 2 (the previous version of Table 2 has now been updated to become Table 3) in the Results section with the differences in PPT tasks expressed as Cohen’s d with 95% confidence interval (CI95) in VR-AOMI (A), AOMI (B) and CTRL (C) groups. We consequently removed redundant data from the main text.
Question: Similar studies conducted in the Discussion section can be presented in a table and discussed.
Answer: We thank the Reviewer for the comment. We apologize for the lack of clarity in specifying the novelty of our study. We have done this also in response to the previous comment (lines 115-121), clarifying that our study was focused on leveraging the embodiment with observed movement induced by AOMI administered through immersive VR, as opposed to AOMI delivered though laptop screen. We also revised the Discussion section to highlight the main result of our research and to elucidate how features of VR administration of AOMI may have facilitated motor learning (lines 493-501 and lines 504-508).
Question: I would also like to point out that the similarity rate in the article is very high.
Answer: We thank the Reviewer for the comment. We have revised and modified a significant portion of the paragraphs within all sections of the manuscript to address this issue.
Reviewer 2 Report
Comments and Suggestions for Authors
The authors of this article present how action observation (AO) in combination with motor imagery (MI) improves manual dexterity in healthy subjects. Furthermore, the improvement is speeded up if the AOMI is combined with an immersive ritual reality. To test their hypothesis, a functional and kinetic evaluation (PPT, NPT and FTT) was applied to a group of healthy volunteers.
The methods are not adequately described.
The results obtained were expected and there was no clear novelty.
Author Response
Bioengineering-2874776 - Revision
The authors express their gratitude to the Editor and the Reviewers for their valuable comments aimed at enhancing the quality of the study and for providing the opportunity to revise the manuscript.
The revised text is titled “Manuscript with track changes” and the sequence of revisions follows the sequence of requested changes.
# Reviewer 2
The authors of this article present how action observation (AO) in combination with motor imagery (MI) improves manual dexterity in healthy subjects. Furthermore, the improvement is speeded up if the AOMI is combined with an immersive ritual reality. To test their hypothesis, a functional and kinetic evaluation (PPT, NPT and FTT) was applied to a group of healthy volunteers.
Question: The methods are not adequately described.
Answer: We thank the Reviewer for the constructive comment. Firstly, we have reorganized the paragraph structure and revised the main text to enhance the clarity of the Methods section (lines 153-334). We have included Figure 1 in paragraph entitled “Intervention” to illustrate the treatment description across the three groups. Additionally, we have inserted Figure 2 in paragraph labeled “Functional and kinematic assessment” to show the assessment timeline in relation to the five-day treatment period.
Question: The results obtained were expected and there was no clear novelty.
Answer: We thank the Reviewer for the comment. We added a paragraph at the end of the Introduction section about the novelty of the study (lines 115-121). Moreover, we substantially revised the Results section to make clearer the main results of our research. Finally, we tried to highlight the main finding and the relevance of our results in the Discussion section (lines 493-515).

Reviewer 3 Report
Comments and Suggestions for Authors
In bioengineering-2874776, Adamo et al. investigated the effects of action observation and motor imagery (AOMI) administered by immersive virtual reality (VR) when compared to AOMI administered through a laptop on hand dexterity.
W1. The authors should proofread their submission.
W2. Some sentences were incomplete. For example, "[19]." is simply a citation by itself, which does not consistute a sentence.
W3. The study was not gender balanced: 15 females vs. 30 males.
W4. The study was also not age balanced: With inclusion criterion of age between 20 and 35 years, the average subject ages were 23.4.
W5. More exhaustive study is needed.
W6. Results are not easy to interpret. More descriptions and discussions are needed.
Comments on the Quality of English Language
W1. The authors should proofread their submission.
W2. Some sentences were incomplete. For example, "[19]." is simply a citation by itself, which does not consistute a sentence.
Author Response
Bioengineering-2874776 - Revision
The authors express their gratitude to the Editor and the Reviewers for their valuable comments aimed at enhancing the quality of the study and for providing the opportunity to revise the manuscript.
The revised text is titled “Manuscript with track changes” and the sequence of revisions follows the sequence of requested changes.
# Reviewer 3
In bioengineering-2874776, Adamo et al. investigated the effects of action observation and motor imagery (AOMI) administered by immersive virtual reality (VR) when compared to AOMI administered through a laptop on hand dexterity.
Question: The authors should proofread their submission.
Answer: We thank the Reviewer for the comment. We apologize for typos and errors in the original manuscript. Two of the authors of the manuscript provided the final revision of the manuscript.
Question: Some sentences were incomplete. For example, "[19]." is simply a citation by itself, which does not constitute a sentence.
Answer: We thank the Reviewer for the comment. We apologize for the error, and we fixed the sentence at line 88. Specifically, the citation [19] was referred to the previous sentence regarding the virtual arm illusion induced by immersive virtual reality.
Question: The study was not gender balanced: 15 females vs. 30 males.
Answer: We thank the Reviewer for the comment. We added this point in the Limitations of the Conclusions section. We discussed this point addressing the fact that the proportion of males to females in the VR-AOMI, AOMI, and CTRL groups did not show statistically significant differences between groups (Table 1).
Question: The study was also not age balanced: With inclusion criterion of age between 20 and 35 years, the average subject ages were 23.4.
Answer: We thank the Reviewer for the comment. We added this point in the Limitations of the Conclusions section. We also specified that the mean age in the three groups did not show statistically significant differences, although there was a difference between the mean age of participants and the mean age computed from the inclusion criteria specified.
Question: More exhaustive study is needed.
Answer: We thank the Reviewer for the comment. To enhance the exhaustiveness of the study, we started clearly specifying the novelty of our study and our hypothesis (lines 115-121). Additionally, we have provided a more detailed description of the study methodology. We tried to clarify further the main findings and the discussion of our results (Results section), establishing links between the main result and existing literature (Discussion section).
Question: Results are not easy to interpret. More descriptions and discussions are needed.
Answer: We thank the Reviewer for the constructive comment. We have re-organized the Results section by making the text more descriptive, renaming subtitles, and removing part of the main text with extensive numerical data. Instead, we have included these data in an additional table in the Results section (new Table 2). Furthermore, we have included Figure 4, which presents a between-group comparison of deltas (expressed as Cohen’s d with a 95% confidence interval) for PPT-L scores, aimed at elucidating the main finding of the study. We also revised the Discussion section to clearly describe the main findings of our study (lines 493-515).

Reviewer 4 Report
Comments and Suggestions for Authors
The paper deals with the problem of hand dexterity improvement of human subjects by employing the virtual reality ability to enhance this process. The proposed approach is interesting, complex and reliable, and the obtained results are useful. The paper is well organised, clearly presented in a logical approach, by using a fine English style. However, the paper must be slightly revised for the sake of clarity and simplicity.
The following issues are recommended to improve the paper:
1. Abstract: state clearly the addressed problem and the novelty of the conducted research.
2. Recommendation to include a Nomenclature for all used acronyms.
3. Introduction: proposal to state here the main contributions and novelty of the paper. Typically, the paper sections are introduced at the end of Introduction.
4. Fix some typing mistakes, e.g., “T2. [37]”, “juggling etc.”, etc. Check carefully the entire manuscript for similar mistakes.
5. Fig. 2: increase the font size, but not exceeding the body text font size. Please comment (give an explanation) on the lower performances of VR-AOMI and AOMI vs. CTRL for PPT-Assembly.
6. The final conclusions are really very general! The Sections 4 and 5 can be reorganised, as typically the conclusions are focused on the main results, the limits of the proposed approach and the future work(s).
Author Response
Bioengineering-2874776 - Revision
The authors express their gratitude to the Editor and the Reviewers for their valuable comments aimed at enhancing the quality of the study and for providing the opportunity to revise the manuscript.
The revised text is titled “Manuscript with track changes” and the sequence of revisions follows the sequence of requested changes.
# Reviewer 4
The paper deals with the problem of hand dexterity improvement of human subjects by employing the virtual reality ability to enhance this process. The proposed approach is interesting, complex and reliable, and the obtained results are useful. The paper is well organised, clearly presented in a logical approach, by using a fine English style. However, the paper must be slightly revised for the sake of clarity and simplicity.
The following issues are recommended to improve the paper:
Question: Abstract: state clearly the addressed problem and the novelty of the conducted research.
Answer: We thank the Reviewer for the comment. We have adjusted the abstract to highlight the novelty of the work and to make clearer the study aim (lines 17-22).
Question: Recommendation to include a Nomenclature for all used acronyms.
Answer: We thank the Reviewer for the comment. We added a paragraph after the Introduction section with the nomenclature of all acronyms used in the manuscript.
Question: Introduction: proposal to state here the main contributions and novelty of the paper. Typically, the paper sections are introduced at the end of Introduction.
Answer: We thank the Reviewer for the constructive comment, Firstly, we added a paragraph in the Introduction section (lines 115-131) to explain the novelty of the study and its contribution to the literature. Secondly, we added a paragraph with an explanation of the paper sections at the end of Introduction section. We provided a brief explanation of each section of the paper to facilitate readers in navigating the content and highlight the relevance of the article (lines 132-139).
Question: Fix some typing mistakes, e.g., “T2. [37]”, “juggling etc.”, etc. Check carefully the entire manuscript for similar mistakes.
Answer: We thank the Reviewer for the comment. We proofread the revised manuscript and corrected mistakes and typos throughout le main text, the tables and figures.
Question: Fig. 2: increase the font size, but not exceeding the body text font size. Please comment (give an explanation) on the lower performances of VR-AOMI and AOMI vs. CTRL for PPT-Assembly.
Answer: We thank the Reviewer for the comment. We increased the font size of Fig.3 (which corresponds to the Fig.2 of the original manuscript). We added a comment on the performances of the three groups in PPT- Assembly task (lines 508-512). In fact, a certain heterogeneity in terms of PPT-Assembly task was found at baseline, where control group revealed higher mean score than AOMI groups. However, this heterogeneity may be attributed to higher variability of the assembly task compared to unimanual tasks during PPT performance in healthy subjects. However, the ANOVA analysis did not reveal any statistically significant between-group differences in the baseline scores of the PPT-Assembly task.
Question: The final conclusions are really very general! The Sections 4 and 5 can be reorganised, as typically the conclusions are focused on the main results, the limits of the proposed approach and the future work(s).
Answer: We thank the Reviewer for the comment. We have reorganized the Discussion and Conclusion sections. In particular, we started the Conclusions section with the main result of the study. We have also addressed the study's limitations and highlighted points for further exploration in future research studies on this topic (lines 575-495).

Round 2
Reviewer 1 Report
Comments and Suggestions for Authors
First of all, thank you for addressing many deficiencies in the revision. I reviewed the updated version of your work again. There are still points that I think are missing in the study. The Abstract section has been updated, but still, no information is provided about the proposed, developed, or used method. The abbreviations presented between lines 129-140 can be given before the references section. There are notes on line 149 and many similar places where the researchers seem to disagree. And these have not been corrected. The expected article is to be more attentive. There is an irrelevant parenthesis on line 279. I think the article was sent too hastily. As a result, the innovations of the study and its contributions to the literature should be at the forefront.
Comments on the Quality of English Language.
Author Response
Bioengineering-2874776 - Revision
The authors express their gratitude to the Editor and the Reviewers for their valuable comments aimed at enhancing the quality of the study and for providing the opportunity to revise the manuscript.
The revised text is titled “Manuscript with track changes” and the sequence of revisions follows the sequence of requested changes.
# Reviewer 1
Quality of English Language
Comment: Minor editing of English language required
Answer: We thank the Reviewer for the comment. One of the authors is an experienced English-speaking colleague who provided English language editing.
Comments and Suggestions for Authors
Comment: First of all, thank you for addressing many deficiencies in the revision. I reviewed the updated version of your work again. There are still points that I think are missing in the study.
Answer: We thank the Reviewer for the general comment on the revised manuscript.
Comment: The Abstract section has been updated, but still, no information is provided about the proposed, developed, or used method.
Answer: We thank the Reviewer for the comment, and we apologize for the lack of clarity in explaining the method used in the study. We detailed study method in the Abstract section at lines 25-30.
Comment: The abbreviations presented between lines 129-140 can be given before the references section.
Answer: We thank the Reviewer for the comment. We moved the Abbreviations section before the References section.
Comment: There are notes on line 149 and many similar places where the researchers seem to disagree. And these have not been corrected. The expected article is to be more attentive.
Answer: We thank the Reviewer for the comment, and we apologize for the oversight. We modified the manuscript at lines 269-270 and line 281 and removed the internal comments in the text. Moreover, we proofread the manuscript and provided a language revision of the revised text.
Comment: There is an irrelevant parenthesis on line 279.
Answer: We thank the Reviewer for the comment, and we apologize for the typos. We removed parenthesis at line 340.
Comment: I think the article was sent too hastily. As a result, the innovations of the study and its contributions to the literature should be at the forefront.
Answer: We thank the Reviewer for the constructive comment. We revised the first part of the Discussion section focusing on the main findings of the study and the contribution to the literature (lines 408-416). Moreover, we edited the Introduction section to underline the innovations of the study (lines 157-160; lines 162-168).
Reviewer 3 Report
Comments and Suggestions for Authors
In bioengineering-2874776R1, Adamo et al. attempted to investigate the effects of action observation and motor imagery (AOMI) administered by immersive virtual reality (VR) when compared to AOMI administered through a laptop on hand dexterity.
W1. The authors should proofread their submission.
W2. The revision contains three (internal) comments.
W3. The authors did not show any information about limb dominance. Any differences between left-handed and right-handed participants.
W4. Some figures are not easy to interpret. For example, Figure 1 shows six subfigures, in which two seem to be identical.
W5. In functional assessment, what kind of raw data were filtered? position or trajectories?
W6. Was the finger tapping test (FTT) a functional or kinematic assessment?
W7. The study was not gender balanced: 15 females vs. 30 males. If interpreted correctly, Table 1 shows no significant differences amongst VR-AOMI, AOMI and CTRL groups. But, it does not show differences between genders.
W8. iThenicate reports show 25% similarity (without bibliography) with 6% from a single source, which appears to be a bit high.
iThenicate reports show 25% similarity (without bibliography) with 6% from a single source, which appears to be a bit high. The authors may consider reducing similarity.
Author Response
Bioengineering-2874776 - Revision
The authors express their gratitude to the Editor and the Reviewers for their valuable comments aimed at enhancing the quality of the study and for providing the opportunity to revise the manuscript.
The revised text is titled “Manuscript with track changes” and the sequence of revisions follows the sequence of requested changes.
# Reviewer 3
Quality of English Language
Comment: Moderate editing of English language required
Answer: We thank the Reviewer for the comment. One of the authors is an experienced English-speaking colleague who provided English language editing.
Comments and Suggestions for Authors
Comment W1: The authors should proofread their submission.
Answer: We thank the Reviewer for the comment, and we apologize for the typos and the lack of clarity of some sentences. Two authors proofread the final version of the revised manuscript and fixed the typos in the article.
Comment W2: The revision contains three (internal) comments.
Answer: We thank the Reviewer for the comment, and we apologize for leaving the comments within the manuscript. We removed all the internal comments and subsequently modified the text to address all points, enhancing the completeness and clarity of the main text (lines 269-270 and line 281).
Comment W3: The authors did not show any information about limb dominance. Any differences between left-handed and right-handed participants.
Answer: We thank the Reviewer for the comment. We have better specified in the inclusion criteria that all participants had right upper limb dominance assessed through Edinburgh Handedness Inventory. We modified the line 210 to increase the clarity of this inclusion criteria.
Comment W4: Some figures are not easy to interpret. For example, Figure 1 shows six subfigures, in which two seem to be identical.
Answer: We thank the Reviewer for the comment. We modified Figure 1, distributing the subfigures differently to enhance the comprehension of the figure and removing the identical subfigure.
Comment W5: In functional assessment, what kind of raw data were filtered? position or trajectories?
Answer: We thank the Reviewer for the comment. We specified that markers trajectories were filtered using a fourth-order low-pass Butterworth filter (cut-off 4 Hz) at line 281.
Comment W6: Was the finger tapping test (FTT) a functional or kinematic assessment?
Answer: We thank the Reviewer for the comment. We inserted the FFT in the kinematic assessment paragraph (lines 269, 270).
Comment W7: The study was not gender balanced: 15 females vs. 30 males. If interpreted correctly, Table 1 shows no significant differences amongst VR-AOMI, AOMI and CTRL groups. But, it does not show differences between genders.
Answer: We thank the Reviewer for the constructive comment. We have clarified in the study limitations that the study was unbalanced in terms of gender, since the M/F ratio was approximately 2:1 regardless of group. However, despite the presence of higher number of male participants, the proportion M/F was constant in each group with no statistical differences, as demonstrated by the p-value of the Chi-squared in Table 1. We have also included the percentage of female participants for each group into Table 1 and revised the text at lines 311,312 to better clarify this concept.
Comment W8: iThenicate reports show 25% similarity (without bibliography) with 6% from a single source, which appears to be a bit high. The authors may consider reducing similarity.
Answer: We thank the Reviewer for the comment. We have extensively revised the Introduction and the first paragraphs of the Discussion sections to overcome this issue.
Round 3
Reviewer 1 Report
Comments and Suggestions for Authors
Thank you for correcting the deficiencies that you did not address in the previous round in this round.
Reviewer 3 Report
Comments and Suggestions for Authors
In bioengineering-2874776R2, Adamo et al. attempted to investigate the effects of action observation and motor imagery (AOMI) administered by immersive virtual reality (VR) when compared to AOMI administered through a laptop on hand dexterity.
S1. The authors addressed Reviewer 3's previous comments.